# Regulation of *KIF2A* by Antitumor *miR-451a* Inhibits Cancer Cell Aggressiveness Features in Lung Squamous Cell Carcinoma

**DOI:** 10.3390/cancers11020258

**Published:** 2019-02-22

**Authors:** Akifumi Uchida, Naohiko Seki, Keiko Mizuno, Yasutaka Yamada, Shunsuke Misono, Hiroki Sanada, Naoko Kikkawa, Tomohiro Kumamoto, Takayuki Suetsugu, Hiromasa Inoue

**Affiliations:** 1Department of Pulmonary Medicine, Graduate School of Medical and Dental Sciences, Kagoshima University, Kagoshima 890-8520, Japan; akiuchi@m3.kufm.kagoshima-u.ac.jp (A.U.); keim@m.kufm.kagoshima-u.ac.jp (K.M.); k8574402@kadai.jp (S.M.); k8173956@kadai.jp (H.S.); kuma@m2.kufm.kagoshima-u.ac.jp (T.K.); taka3741@m2.kufm.kagoshima-u.ac.jp (T.S.); inoue-pulm@umin.net (H.I.); 2Department of Functional Genomics, Graduate School of Medicine, Chiba University, Chuo-ku, Chiba 260-8670, Japan; yasutaka1205@olive.plala.or.jp (Y.Y.); naoko-k@hospital.chiba-u.jp (N.K.)

**Keywords:** microRNA, *miR-451a*, *KIF2A*, lung squamous cell carcinoma, antitumor

## Abstract

In the human genome, *miR-451a* is encoded close to the *miR-144* on chromosome region 17q11.2. Our previous study showed that both strands of pre-*miR-144* acted as antitumor miRNAs and were involved in lung squamous cell carcinoma (LUSQ) pathogenesis. Here, we aimed to investigate the functional significance of *miR-451a* and to identify its targeting of oncogenic genes in LUSQ cells. Downregulation of *miR-451a* was confirmed in LUSQ clinical specimens, and low expression of *miR-451a* was significantly associated with poor prognosis of LUSQ patients (overall survival: *p* = 0.035, disease-free survival: *p* = 0.029). Additionally, we showed that ectopic expression of *miR-451a* significantly blocked cancer cell aggressiveness. In total, 15 putative oncogenic genes were shown to be regulated by *miR-451a* in LUSQ cells. Among these targets, high kinesin family member 2A (*KIF2A*) expression was significantly associated with poor prognosis (overall survival: *p* = 0.043, disease-free survival: *p* = 0.028). Multivariate analysis showed that *KIF2A* expression was an independent prognostic factor in patients with LUSQ (hazard ratio = 1.493, *p* = 0.034). Aberrant *KIF2A* expression promoted the malignant transformation of this disease. Analytic strategies based on antitumor miRNAs and their target oncogenes are effective tools for identification of novel molecular pathogenesis of LUSQ.

## 1. Introduction

Lung cancer is the most common cause of cancer-related death worldwide, accounting for more than 1.7 million deaths each year [1]. The most common type of lung cancer is non-small cell lung cancer (NSCLC), which can be divided into several subtypes, including squamous cell carcinoma (LUSQ), adenocarcinoma (LUAD), and large cell carcinoma [2]. Patients with LUAD show improved survival rates following treatment with epidermal growth factor receptor tyrosine kinase inhibitors, anaplastic lymphoma kinase tyrosine kinase inhibitors, and immune checkpoint inhibitors [3,4,5,6].

LUSQ remains a common cancer among NSCLCs, and over 400,000 people worldwide are diagnosed with LUSQ each year [7]. The majority of patients with LUSQ have a history of heavy smoking, highlighting tobacco-related carcinogenesis as a clear causative factor of LUSQ. Many mutations accumulate in LUSQ cells owing to the influence of heavy smoking over many years [7]. However, no universal therapeutic targets have currently been found for LUSQ, despite many studies. In contrast to treatment for LUAD, conventional platinum-based chemotherapy has still been performed for inoperable cases of LUSQ in the past two decades [7].

Furthermore, given the malignant nature of LUSQ, it has high invasive and metastatic potential. Distant metastases at the time of presentation of LUSQ are a frequent clinical problem. Many patients with LUSQ present with metastatic disease at the time of diagnosis [8]. For these reasons, continued research, applying advanced genomic-based approaches, is indispensable for identification of novel biomarkers for earlier detection and for development of effective targeted molecular therapies for LUSQ.

MicroRNAs (miRNAs) are small noncoding RNAs (19–24 nucleotides in length) that modulate the expression of many genes by blocking translation or degrading mRNAs in a sequence-dependent manner [9]. Notably, one miRNA can regulate the expression of many protein-coding and noncoding RNA transcripts [10]. Thus, aberrantly expressed miRNAs can disrupt normal cell function, thereby supporting cancer pathogenesis [11]. Many studies have shown that aberrantly expressed miRNAs are involved in the pathogenesis of many diseases, including cancer [12,13,14].

It is possible to explore oncogenic networks in LUSQ cells controlled by oncogenic or antitumor miRNAs using advanced genomic approaches. We have sequentially identified antitumor miRNAs and their targeted oncogenic genes and pathways in LUSQ cells, e.g., *miR-145-5p*/*-3p* (targeting oncogene: *MTDH*), *miR-150-5p* (*MMP14*), *miR-29*-family (*LOXL2*), and *miR-218* (*TPD52*) [15,16,17,18]. Recently, we revealed that both strands of *miR-144* (*miR-144-5p* (the passenger strand) and *miR-144-3p* (the guide strand)) act as antitumor miRNAs and that these miRNAs significantly block malignant abilities through coordinated targeting of *NCS1* [19]. Furthermore, analysis of the expression profiles of *miR-144-5p*, *miR-144-3p*, and *NCS1* can be used to help predict prognosis in patients with LUSQ [19]. Researchers are now recognizing miRNA passenger strands as active players in cancer pathogenesis.

In this study, we focused on *miR-451a* because it has been shown to form miRNA clusters (*miR-144-5p*/*miR-144-3p*/*miR-451a*) located in the human chromosome 17q11.2 region. Downregulation of *miR-451a* was confirmed in LUSQ clinical specimens, and low expression of *miR-451a* was found to be significantly associated with poor prognosis in patients with LUSQ (overall survival (OS): *p* = 0.035, disease-free survival (DFS): *p* = 0.029). We investigated the functional significance of *miR-451a* in LUSQ cells and identified the oncogenic genes regulated by *miR-451a* in LUSQ pathogenesis. Moreover, kinesin family member 2A (*KIF2A*) was directly controlled by *miR-451a* and its expression was closely associated with LUSQ pathogenesis. Analytic strategies based on antitumor miRNAs and their target oncogenes are effective tools for identification of novel molecular pathogenesis of LUSQ.

## 2. Results

### 2.1. Downregulation of miR-451a in LUSQ Clinical Specimens and Its Clinical Significance

In total, 50 clinical specimens (30 LUSQ tissues and 20 noncancerous lung tissues) were obtained from patients who underwent thoracic surgery at Kagoshima University Hospital. The characteristics of the patients are shown in Table 1. The expression level of *miR-451a* was significantly downregulated in LUSQ tissues as compared with those in noncancerous tissues (*p* < 0.001, Figure 1A). In two LUSQ cell lines, EBC-1 and SK-MES-1, the expression levels of *miR-451a* were markedly low (Figure 1A).

To investigate the clinical significance of *miR-451a* in LUSQ, we applied The Cancer Genome Atlas (TCGA) database analyses. Patients with low expression of *miR-451a* showed significantly poor prognosis compared with patients with high expression of *miR-451a* (5-year OS: *p* = 0.035 and 5-year DFS: *p* = 0.029, Figure 1B). Furthermore, in LUSQ patients with adjusting clinical stage and age distribution, low expression of *miR-451a* also predicted poor prognosis compared with high expression of *miR-451a* (5-year OS: *p* = 0.026 and 5-year DFS: *p* = 0.024, Appendix A).

Multivariate analysis showed that low expression of *miR-451a* was an independent prognostic factor in patients with LUSQ (hazard ratio = 0.667, *p* = 0.029, Figure 1D). By analyzing combination *miR-451a*, *miR-144-3p* and *miR-144-5p* expression, combination both high expression of *miR-451a* and *miR-144-5p* predicted additive poor prognosis compared with high expression *miR-451a* alone or *miR-144-5p* alone (Appendix A).

In addition, TCGA database analyses showed that low expression of *miR-451a* was associated with poor prognosis in patients with renal papillary cell carcinoma and renal clear cell carcinoma (Appendix A).

### 2.2. Induction of Apoptotic Cells by Ectopic Expression of miR-451a in LUSQ Cells

First, we investigated the antitumor roles of *miR-451a* in LUSQ cells using ectopic expression of mature miRNAs in EBC-1 and SK-MES-1 cells. Cell proliferation assays indicated significant inhibition of cell growth in *miR-451a*-transfected cells compared with that in mock- or control-transfected cells (Figure 2A,D).

We further investigated the occurrence of apoptosis using flow cytometry assays. Our data showed that the percentages of apoptotic cells were increased in *miR-451a*-transfected cells in comparison with those in mock- or control-transfected cells (Figure 2B,C,E,F). Moreover, restoration of *miR-451a* expression promoted cleaved poly (ADP-ribose) polymerase (PARP) expression (Figure 3).

### 2.3. Effects of Ectopic Expression of miR-451a on LUSQ Cell Migration and Invasion

We then investigated the potential effects of *miR-451a* on cell migration and invasion in LUSQ cells. Overexpression of *miR-451a* attenuated cancer cell migration and invasion in comparison with that in mock- or control-transfected cells (Figure 4A,B).

### 2.4. Screening of Putative Target Genes by miR-451a Regulation in LUSQ Cells

Next, we aimed to identify putative target genes of *miR-451a*. To this end, we performed miRNA database analyses and comprehensive gene expression assays. The search strategy for miRNA targets is presented in Appendix A.

Using TargetScanHuman database (release 7.2), we found that 548 putative target genes had binding sites for *miR-451a* in their 3′-untranslated regions (UTRs). We then selected genes that showed increased expression in NSCLC specimens (Gene Expression Omnibus (GEO) accession number: GSE19188), merged gene expression analysis data with *miR-451a*-transfected SK-MES-1 cells (GEO accession number: GSE113066), and selected genes showing decreased expression. After this analysis, 15 candidate *miR-451a* target genes were identified (Table 2).

We then investigated the clinical impact of these target genes in LUSQ using TCGA database analyses. Among these candidate genes, high expression of *KIF2A* was significantly associated with LUSQ pathogenesis (5-year OS: *p* = 0.043 and 5-year DFS: *p* = 0.028; Figure 5). Furthermore, among patients with adjusting clinical stage and age distribution, patients with high expression of *KIF2A* also showed significantly poor prognosis compared with patients with low expression of *KIF2A* (5-year OS: *p* = 0.029; Appendix A). Moreover, among LUSQ patients with early clinical stage (stage I and II), high expression of *KIF2A* was significantly associated not only with prognosis but also with cancer recurrence (Appendix A). Therefore, we focused on *KIF2A* and validated the functional implications in LUSQ cells.

In addition, TCGA database analyses showed that high expression of *KIF2A* was associated with poor prognosis in patients with renal papillary cell carcinoma and hepatocellular carcinoma (Appendix A).

### 2.5. Expression of KIF2A Was Directly Controlled by miR-451a in LUSQ Cells

Next, we examined the control of *KIF2A* expression by *miR-451a*. Levels of KIF2A mRNA and protein were significantly suppressed by *miR-451a* transfection into EBC-1 and SK-MES-1 cells compared with those in mock- or control-transfected cells (Figure 6A,B).

Luciferase assays were then used to confirm the direct binding of *miR-451a* to *KIF2A* mRNA. Based on an analysis of the TargetScanHuman database (Release 7.2), there was a putative binding site for *miR-451a* in the 3′-UTR of *KIF2A* (position 52–58, Figure 6C). Accordingly, luciferase reporter assays were performed using a vector harboring these sequences in order to determine whether *miR-451a* directly regulated *KIF2A* expression in a sequence-dependent manner.

We observed greatly reduced luminescence after transfection with *miR-451a* and the vector carrying the wild-type 3′-UTR of *KIF2A*. Transfection with the deletion-type vector did not reduce luminescence intensities in EBC-1 or SK-MES-1 cells (Figure 6C). These findings demonstrated that *miR-451a* bound directly to the 3′-UTR of *KIF2A*.

We also investigated the correlation between *miR-451a* and *KIF2A* expression in LUSQ patients. TCGA database analyses showed that a negative correlation was detected between *miR-451a* and *KIF2A* expression in LUSQ patients (*r* = −0.180 and *p* = 0.010; Appendix A).

### 2.6. Aberrant Expression of KIF2A and Its Clinical Significance in LUSQ

We then validated the expression of KIF2A protein in LUSQ clinical tissues using immunohistochemical analyses. Compared with normal lung specimens, KIF2A protein was strongly expressed in LUSQ tissues (Figure 7A–D and Table 3).

Moreover, multivariate analysis showed that *KIF2A* overexpression was an independent predictive factor for OS (hazard ratio = 1.493, *p* = 0.034; Figure 8).

### 2.7. Effects of KIF2A Knockdown on Cell Proliferation and Induction of Apoptotic Cells in LUSQ Cells

To further confirm the role of *KIF2A* in the pathogenesis of LUSQ, we next evaluated the effects of *KIF2A* downregulation in EBC-1 and SK-MES-1 cells using small interfering RNAs (siRNAs). Both si-*KIF2A*-1 and si-*KIF2A*-2 effectively decreased the expression of KIF2A mRNA and protein (Figure 9A,B).

Cancer cell proliferation was significantly suppressed by si-*KIF2A* transfection in comparison with those in mock- or control-transfected LUSQ cells (Figure 10A,D).

Moreover, the apoptotic cell numbers were increased in si-*KIF2A* transfected cells compared with those in mock- or control-transfected cells (Figure 10B,C,E,F). Cleaved PARP expression was also detected in si-*KIF2A*-transfected cells (Figure 11).

### 2.8. Effects of KIF2A Silencing on Cancer Cell Migration and Invasion in LUSQ Cells

Further analyses showed that cancer cell motility, including migration and invasive abilities, was markedly inhibited by knockdown of *KIF2A* via si-*KIF2A* transfection compared with those in mock- or control-transfected LUSQ cells (Figure 12A,B).

### 2.9. Identification of KIF2A-Mediated Downstream Pathways in LUSQ Cells

Based on our above findings, we then evaluated the downstream genes regulated by *KIF2A* using genome-wide gene expression analyses and in silico analyses in si-*KIF2A*-transfected SK-MES-1 cells. Our strategy is shown in Appendix A. In total, 3621 genes were identified as downregulated genes in si-*KIF2A*-transfected cells compared with that in mock-transfected cells (GEO accession number: GSE123318). Of these 3621 genes, 92 genes were upregulated in NSCLC clinical specimens (GEO accession number: GSE19188).

As a result of classifying 92 genes into KEGG pathways, 5 pathways were identified, including cell cycle, p53 signaling pathway and cell cycle, p53 signaling pathway, DNA replication, and pathways in cancer (Table 4 and Table 5).

## 3. Discussion

In the human genome, several miRNAs encoded in close proximity within a chromosome region are defined as clustered miRNAs. Analyses of our original miRNA signatures by RNA sequencing revealed that several miRNA clusters, including *miR-1*/*miR-133a*, *miR-206*/*miR-133b*, *miR-23b*/*miR-27b*/*miR-24-1*, *miR-143*/*miR-145*, and *miR-221*/*miR-222*, are frequently downregulated in several types of cancer tissues [20,21,22,23,24,25,26,27]. Our previous studies showed that these miRNA clusters act as antitumor miRNAs by targeting several oncogenic genes [20,21,22,23,24,25,26,27]. In LUSQ cells, *miR-1* and *miR-133a* are significantly downregulated in LUSQ tissues, and ectopic expression of these miRNAs inhibits cancer cell malignant phenotypes [20]. Furthermore, *CORO1C* is coordinately regulated by *miR-1* and *miR-133a*, and aberrant expression of *CORO1C* enhances the migration and invasive abilities of LUSQ cells [20].

Notably, our miRNA signatures demonstrated that *miR-144-5p*, *miR-144-3p,* and *miR-451a* frequently showed decreased expression levels in several cancers, including head and neck cancer, renal cell carcinoma, and bladder cancer. Moreover, these three miRNAs were found to form a miRNA cluster in human chromosome 17q11.2 [28,29,30,31]. Our previous studies revealed that the members of this miRNA cluster act as antitumor miRNAs by targeting oncogenic genes, including *CCNE1*, *CCNE2*, *ESDN*/*DCBLD2*, *SDC3*, and *PMM2* [28,29,30,31]. Recently, we also demonstrated that *miR-144-5p* and *miR-144-3p* have antitumor functions in LUSQ cells and that their expression predicts a poor prognosis [19]. Interestingly, in this study, *miR-451a* also acts as an antitumor miRNA, and low *miR-451a* expression predicts poor prognosis for patients with LUSQ. Based on our previous studies and current data, we have concluded that members of this miRNA cluster are closely involved in LUSQ pathogenesis.

Downregulation of *miR-451a* has been reported in other types of cancers and its ectopic expression inhibits cancer cell aggressiveness, including that in gastric cancer, glioblastoma, nasopharyngeal cancer, renal cell carcinoma, and prostate cancer [28,30,32,33,34,35]. In LUAD cells, ectopic expression of *miR-451a* inhibits cancer cell proliferation and enhances apoptosis via targeting of *RAB14* and AKT signaling pathways [36]. Furthermore, downregulation of *miR-451* was detected in NSCLC tissues and its expression was an independent predictor of prognosis of NSCLC, such as advanced disease stage and metastasis [37]. Interestingly, ectopic expression of *miR-451* suppressed cell proliferation, migration and activation of AKT through targeting *MIF* in NSCLC cells [37]. Based on these facts, we concluded that *miR-451a* is a pivotal antitumor miRNA in human cancers through targeting of several oncogenic genes.

Elucidation of molecular mechanisms of aberrantly expressed miRNAs in cancer cells is an important issue for cancer research. Previous study showed that expression of *miR-451a* was significantly recovered by treatment with 5-aza-2′-deoxycitidine or sodium plenylbutyrate in NSCLC cells [36]. These results indicated that DNA hypermethylation was caused to downregulation of *miR-451a* in NSCLC. Recent study of prostate cancer showed that *HP1γ* was upregulated by oncogenic *c-MYC* and *HP1γ* suppressed to expression of *miR-451a* in prostate cancer cells [38]. Further investigation of the molecular mechanism of downregulation of *miR-451a* in LUSQ cells is indispensable.

As a unique natural feature of miRNAs, a single miRNA can regulate vast numbers of RNA transcripts. The RNA transcripts controlled by miRNA vary depending on the cell type. Therefore, the next task is to find the oncogenic genes and pathways that are controlled by antitumor *miR-451a* in LUSQ cells. In total, 15 putative oncogenic targets under *miR-451a* regulation were successfully identified in this study. Among these targets, *MIF* was reported by another group as a direct target of *miR-451a*, and its expression was shown to enhance cancer cell growth and invasion in gastric cancer and nasopharyngeal cancer [32,34]. Understanding the targets that are regulated by the *miR-451a* in LUSQ cells may contribute to our knowledge of LUSQ molecular pathogenesis.

In this study, we focused on *KIF2A* because its expression was significantly associated with a worse prognosis for patients with LUSQ. Our current data confirmed that aberrant expression of *KIF2A* enhanced cancer cell aggressiveness in LUSQ cells. *KIF2A* is a member of the kinesin-13 family (other members are *KIF2B*, *KIF2C*, and *KIF24*), and its main function is microtubule depolymerization, a critical event for mitotic progression and spindle assembly [39,40,41]. Recently, several reports have shown that aberrantly expressed *KIF2A* is detected in several cancers, including breast, oral, colorectal, and ovarian cancers, and its expression contributes to cancer cell malignancies [42,43,44,45]. In a recent study, expression of *KIF2A* was found to be closely associated with TNM stage and lymph node metastasis in LUAD [46]. Knockdown of *KIF2A* in LUAD cells inhibits cell proliferation and induces apoptosis [46], consistent with our current findings of LUSQ cells. In lung cancers (LUSQ and LUAD), aberrantly expressed *KIF2A* may serve as a valuable prognosis marker and candidate for therapeutic targeting.

Furthermore, to investigate *KIF2A*-mediated oncogenic pathways in LUSQ cells, we applied genome-wide gene expression analyses and in silico analyses using *KIF2A*-knockdown LUSQ cells. We identified several *KIF2A*-mediated pathways, including cell cycle, p53 signaling, and DNA replication. Among the genes involved in these pathways, a previous study showed that overexpression of *MCM2* is associated with OS in patients with LUSQ and that its aberrant expression participates in the development and progression of LUSQ [47]. Moreover, *MCM4* was detected in clinical specimens of LUSQ [48]. Aberrant expression of these oncogenic genes contributes to the development, metastasis, and drug resistance of LUSQ. Elucidation of novel RNA networks controlled by antitumor miRNAs will accelerate the journey to a comprehensive understanding of LUSQ molecular pathogenesis.

## 4. Materials and Methods

### 4.1. Clinical Samples Collection, Cell Lines, and RNA Extraction

The current study was approved by the Bioethics Committee of Kagoshima University Hospital (Kagoshima, Japan; approval numbers: 26-164). Prior written informed consent and approval for this study were obtained from each patient. We collected 50 lung samples at Kagoshima University Hospital from 2010 to 2013. The pathological stages of LUSQ were classified according to the International Association for the Study of Lung Cancer TNM classification, 7th Edition [49]. Lung cancer samples and noncancerous tissues were obtained from the lung specimens resected by thoracic surgery for LUSQ. The extraction of RNA from formalin-fixed, paraffin-embedded specimens was performed as described in a previous study [17].

Two LUSQ cell lines, EBC-1 and SK-MES-1, were obtained from the Japanese Cancer Research Resources Bank (Osaka, Japan) and the American Type Culture Collection (ATCC; Manassas, VA, USA), respectively. The procedures for cell culture, extraction of total RNA, and extraction of protein from LUSQ cell lines were described in our earlier manuscripts [17,18,50].

### 4.2. Quantitative Real-Time Reverse Transcription Polymerase Chain Reaction (qRT-PCR)

The methods for qRT-PCR have been described previously [16,50,51]. TaqMan qRT-PCR assays (assay ID: 001141; Applied Biosystems, Foster City, CA, USA) were used to validate *miR-451a* expression. To normalize the data, the expression of *RNU48* (assay ID: 001006; Applied Biosystems) was used. *KIF2A* expression values were measured using TaqMan probes and primers (assay ID: Hs00189636_m1; Applied Biosystems). Expression of glyceraldehyde 3-phosphate dehydrogenase (*GAPDH*; assay ID: Hs99999905_m1; Applied Biosystems) was used for normalization.

### 4.3. Transfections with Mature miRNA and siRNA into LUSQ Cell Lines

We used the following mature miRNA species and siRNAs in this study: mirVana miRNA mimic, hsa*-miR-451a* (product ID: MC10286; Applied Biosystems), and Stealth Select RNAi siRNA, si-*KIF2A* (P/N: HSS105799 and HSS180178; Invitrogen, Carlsbad, CA, USA). Anti-miR Negative Control #1 (catalog no: AM17010; Applied Biosystems) was used as a negative control. The method for transfection is described in our previous studies [15,16,50,52].

### 4.4. Cell proliferation, Migration, and Invasion Assays

Cell proliferation activity was determined by XTT assays using a Cell Proliferation Kit (Biological Industries, Beit-Haemek, Israel). Cell migration ability was evaluated with wound healing assays. Cell invasion ability was determined with Corning Matrigel Invasion Chambers (Discovery Labware, Inc., Bedford, MA, USA). Detailed procedures were described in our earlier reports [16,20].

### 4.5. Apoptosis Assays

Apoptotic cells were detected using a FITC Annexin V Apoptosis Detection Kit (BD Biosciences, Bedford, MA, USA) according to the manufacturer’s protocol and analyzed by BD FACS Celesta Flow Cytometer (BD Biosciences). Cells were identified as viable cells, dead cells, and early and late apoptotic cells, and the percentages of apoptotic cells under different experimental conditions were compared [53]. We used 15 μM cisplatin as a positive control.

### 4.6. Identification of Putative Target Genes Regulated by miR-451a in LUSQ Cells

The strategy for selecting target genes in this study is shown in Appendix A. We selected putative target genes having binding sites for *miR-451a* using TargetScanHuman ver.7.2 (http://www.targetscan.org/vert_72/) (data was downloaded on 13 July 2018). We then examined the expression levels of putative *miR-451a* targets in the NSCLC clinical expression data from the GEO database (GSE19188). The microarray data were deposited in the GEO repository under accession number GSE113066.

### 4.7. Clinical Database Analysis

The following databases were applied to investigate the clinical significance of gene expression in patients with LUSQ: TCGA database (https://tcga-data.nci.nih.gov/tcga/), cBioPortal (http://www.cbioportal.org/), and OncoLnc (http://www.oncolnc.org/) (data was downloaded on August 8, 2018) [54,55]. Lower and upper percentiles of TCGA database were both 33, without additional comments.

### 4.8. Plasmid Construction and Dual-Luciferase Reporter Assay

Partial sequences of the wild-type *KIF2A* 3′-UTR, either containing or lacking the *miR-451a* target site, were cloned into the psiCHECK-2 vector (C8021; Promega, Madison, WI, USA).

After cotransfecting miRNA and the constructed vector into EBC-1 and SK-MES-1 cells, firefly and *Renilla* luciferase activities were determined using a Dual-Luciferase Reporter Assay System (catalog no: E1960; Promega). The procedure is described in our previous studies [15,16,50,52,56].

### 4.9. Western Blotting and Immunohistochemistry

Membranes were immunoblotted with rabbit polyclonal anti-KIF2A antibodies (1:1000 dilution; ab197988; Abcam, Cambridge, UK) and monoclonal anti-GAPDH antibodies (1:20000 dilution; MAB374; EMD Millipore, Billerica, MA, USA). The procedures for western blotting were as described previously [16,18].

Immunohistochemistry was performed with a VECTASTAIN Universal Elite ABC Kit (catalog no: PK-6200; Vector Laboratories, Burlingame, CA, USA) according to the manufacturer’s protocol. The characteristics of patients included in the tissue microarray (catalog no: LC813a; US Biomax, Inc., Derwood, MD, USA) are shown in Table 3. The procedure for immunohistochemistry is described in our earlier reports [16,21,50].

### 4.10. Identification of Downstream Targets Regulated by KIF2A in LUSQ Cells

The microarray expression profiles of si-*KIF2A* transfectants were deposited in the GEO repository under accession number GSE123318. KEGG pathway categories, determined with the GeneCodis program (http://genecodis.cnb.csic.es/), were used to reveal the signaling pathways regulated by *KIF2A* [57]. The strategy for identification of signaling pathways is shown in Appendix A.

### 4.11. Statistical Analysis

All data were analyzed using SPSS version 25 software (IBM SPSS, Chicago, IL, USA). To assess the significance of differences between 2 groups, we used Mann-Whitney *U* tests. Differences between multiple groups were examined by one-way analysis of variance and Tukey tests for post-hoc analysis. We used Kaplan–Meier survival curves and log-rank statistics to analyze the differences between OS rates and DFS rates. All patients with identifying the period of DFS among OS analysis were used for analysis of DFS rate. To adjust clinical stage and age distribution in TCGA database, we used propensity score matching analysis using a multivariable logistic regression model, and one-to-one pair matching was carried out without replacement. Correlations were examined using Spearman’s correlation test. Univariate and multivariate analyses for 5-year OS using TCGA database were carried out by Cox proportional hazards regression analyses.

## 5. Conclusions

In this study, our results showed that *miR-451a* was significantly downregulated in LUSQ tissues. Moreover, we found that this miRNA acted as a tumor suppressor in LUSQ cells and directly regulated *KIF2A*. Functional analyses showed that *KIF2A* was a significant gene in LUSQ pathogenesis and that overexpression of *KIF2A* was involved in the pathogenesis of LUSQ, thereby characterizing *KIF2A* as an oncogene. Our approach, to identify aberrantly expressed miRNAs and their downstream cancer-related genes, is a groundbreaking strategy to uncover the novel molecular mechanisms mediating the pathogenesis of LUSQ.

## Figures and Tables

**Figure 1 cancers-11-00258-f001:**
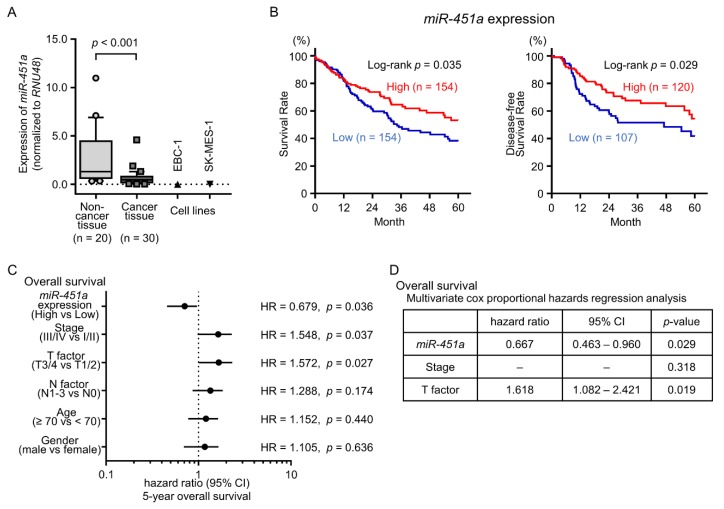
Expression levels of *miR-451a* in lung squamous cell carcinoma (LUSQ) clinical specimens and association with prognosis in patients with LUSQ. (**A**) *miR-451a* expression levels in clinical specimens and cell lines (EBC-1 and SK-MES-1). (**B**) Kaplan–Meier curve of 5-year overall survival and 5-year disease-free survival according to *miR-451a* expression among patients with LUSQ in The Cancer Genome Atlas (TCGA) database (*p* = 0.035 and *p* = 0.029, respectively). Patients were divided into high (red) and low (blue) expression groups. (**C**,**D**) Forest plot of univariate Cox proportional hazards regression analysis and multivariate Cox proportional hazards regression analysis of 5-year overall survival for *miR-451a* expression using TCGA database.

**Figure 2 cancers-11-00258-f002:**
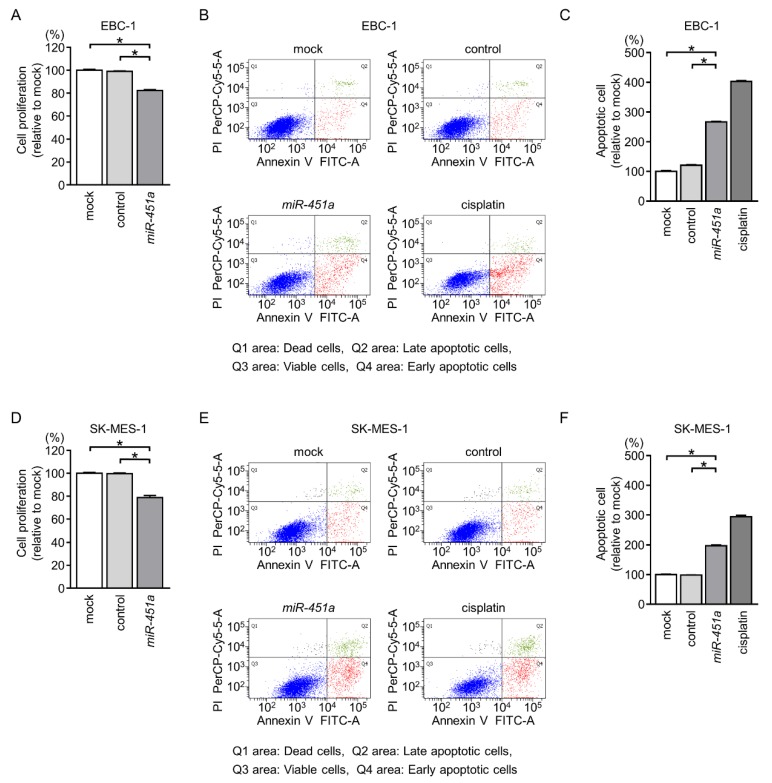
Cell proliferation and apoptosis assays following ectopic expression of *miR-451a* in LUSQ cells. (**A**,**D**) Cell proliferation was determined by XTT assays 72 h after transfection with *miR-451a* (* *p* < 0.001). (**B**,**E**) Apoptosis assays using flow cytometry with Annexin V-FITC- and PI-PerCP-Cy5-5-A-stained cells. Cisplatin (15 μM) was used as a positive control for induction of apoptosis. (**C**,**F**) Quantification of apoptotic cells following ectopic expression of *miR-451a* in LUSQ cells (EBC-1 and SK-MES-1). The normalized ratios of apoptotic cells are shown as histograms from FACS analyses (* *p* < 0.001).

**Figure 3 cancers-11-00258-f003:**
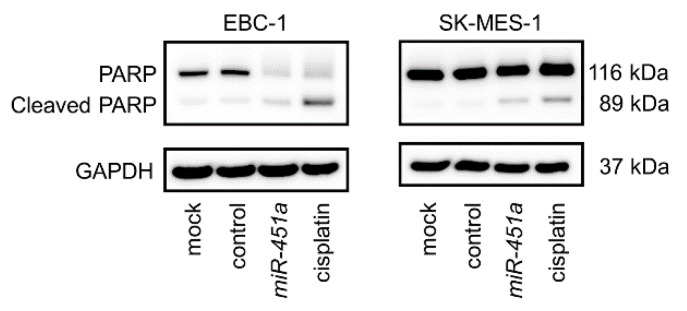
Western blot analyses of cleaved polymerase (PARP) as a marker of apoptosis in LUSQ cell lines. glyceraldehyde 3-phosphate dehydrogenase (GAPDH) was used as a loading control.

**Figure 4 cancers-11-00258-f004:**
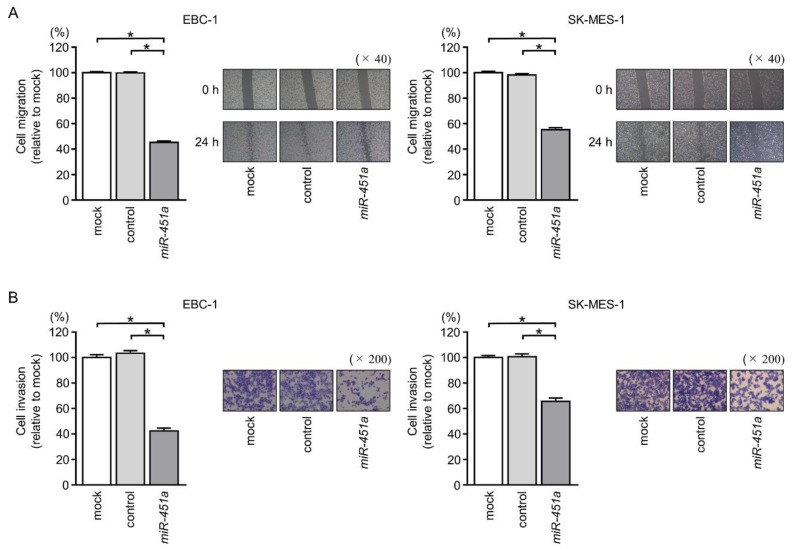
Cell migration and invasion assays following ectopic expression of *miR-451a* in LUSQ cells. (**A**) Cell migration was measured by wound healing assays (**p* < 0.001). (**B**) Cell invasion was determined by Matrigel invasion assays (**p* < 0.001). Phase-contrast micrographs of LUSQ cells in migration and micrographs of LUSQ cells in invasion assays are shown.

**Figure 5 cancers-11-00258-f005:**
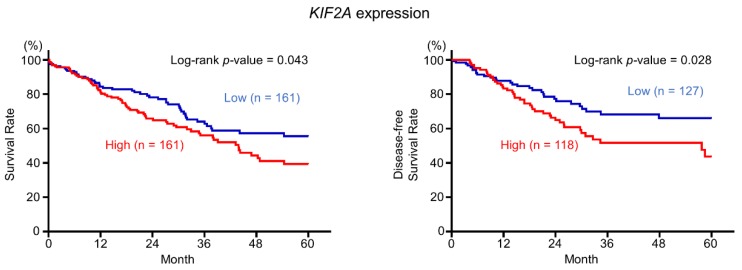
Kaplan–Meier analyses of *KIF2A* expression. High *KIF2A* expression group (red line); low *KIF2A* expression group (blue line).

**Figure 6 cancers-11-00258-f006:**
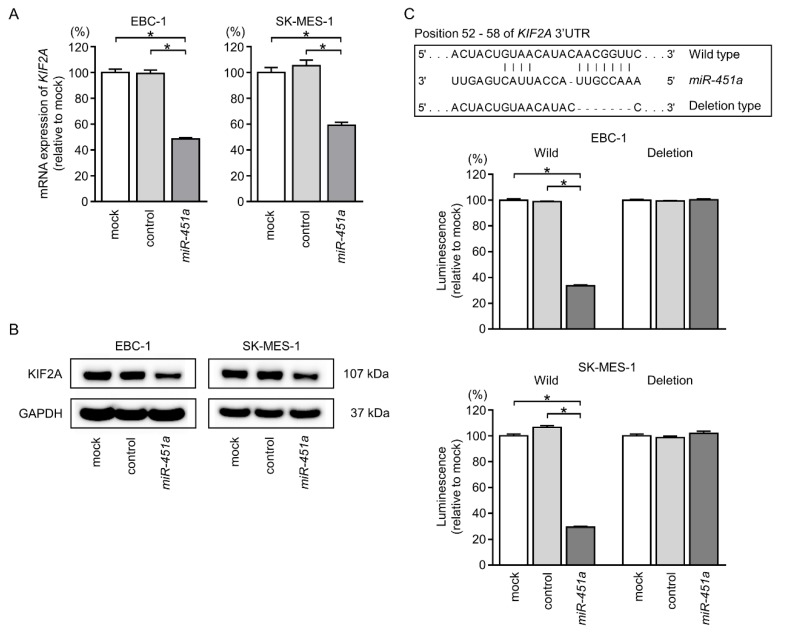
Direct regulation of *KIF2A* expression by *miR-451a* in LUSQ cells. (**A**) *KIF2A* mRNA expression levels 72 h after transfection of EBC-1 or SK-MES-1 cells with 10 nM *miR-451a* (* *p* < 0.001). (**B**) Protein expression of KIF2A 72 h after transfection with *miR-451a*. (**C**, upper) Putative *miR-451a* binding site in the 3′-UTR of *KIF2A* mRNA. (**C**, lower) Dual luciferase reporter assays using vectors encoding putative *miR-451a* target sites in the *KIF2A* 3′-UTR (position 52–58) for both wild-type and deleted regions. Normalized data were calculated as *Renilla*/firefly luciferase activity ratios (* *p* < 0.001).

**Figure 7 cancers-11-00258-f007:**
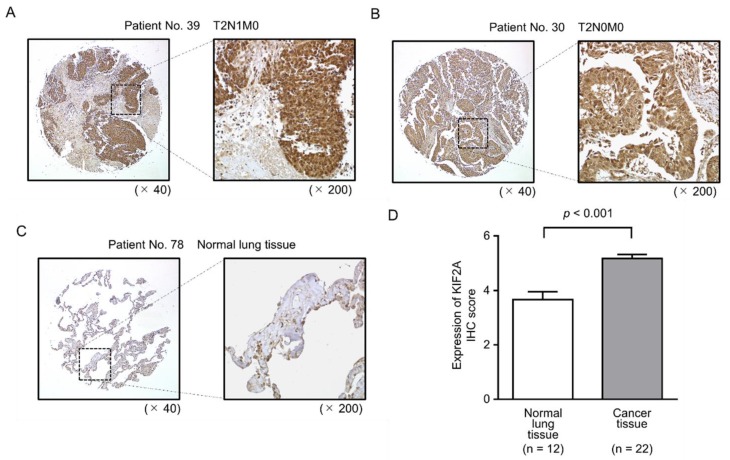
Aberrant expression of KIF2A in clinical LUSQ specimens. (**A**–**C**) Aberrant expression of KIF2A was observed in the cytoplasm of cancer cells, whereas negative or low expression of KIF2A was observed in normal cells. (**D**) Comparison of KIF2A expression scoring across clinical lung specimens. KIF2A expression in LUSQ tissues was significantly higher than that in normal lung tissue (* *p* < 0.001).

**Figure 8 cancers-11-00258-f008:**
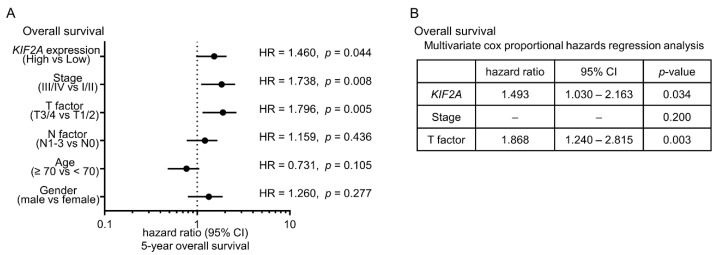
Clinical significance of *KIF2A* expression in patients with LUSQ. (**A**) Forest plot of univariate Cox proportional hazards regression analysis of 5-year overall survival. (**B**) Multivariate Cox proportional hazards regression analysis of 5-year overall survival using TCGA database.

**Figure 9 cancers-11-00258-f009:**
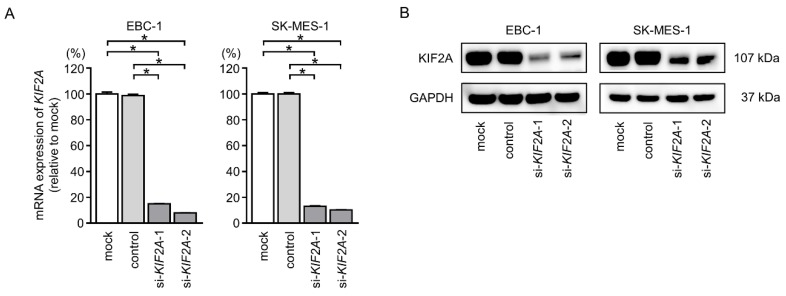
Efficiency of *KIF2A* silencing by siRNA transfection. (**A**) mRNA expression of *KIF2A* 72 h after transfection with si-*KIF2A* in EBC-1 and SK-MES-1 cells. (* *p* < 0.001). (**B**) KIF2A protein expression was determined by Western blot analysis 72 h after transfection with si-*KIF2A* in EBC-1 and SK-MES-1 cells.

**Figure 10 cancers-11-00258-f010:**
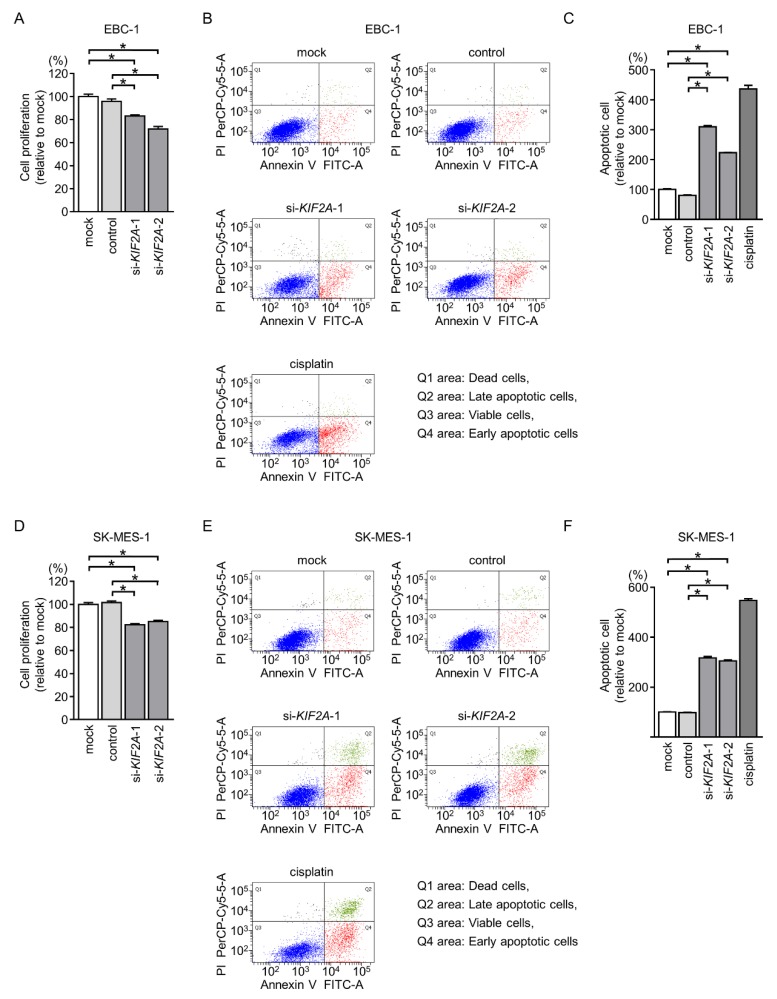
Inhibition of cell proliferation and induction of apoptotic cells by silencing of *KIF2A* expression in LUSQ cells. (**A**,**D**) Cell proliferation was identified by XTT assays 72 h after transfection with si-*KIF2A*-1 and si-*KIF2A*-2 (* *p* < 0.001). (**B**,**E**) Apoptosis was determined by flow cytometry. Cisplatin (15 μM) was used as a positive control for induction of apoptosis. (**C**,**F**) Quantification of apoptotic cells by silencing of *KIF2A* expression in LUSQ cells (EBC-1 and SK-MES-1). The normalized ratios of apoptotic cells are shown as histograms from FACS analyses (* *p* < 0.001).

**Figure 11 cancers-11-00258-f011:**
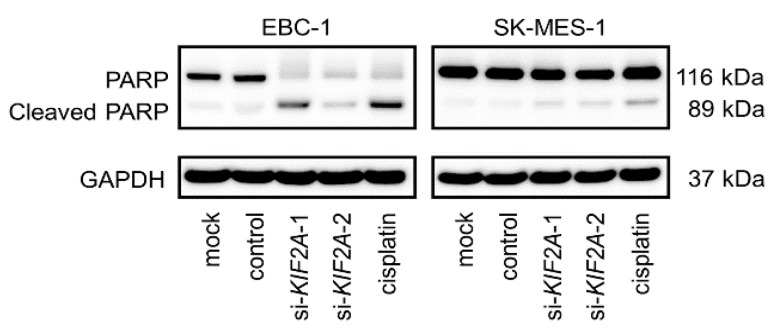
Western blot analyses of cleaved PARP as a marker of apoptosis in LUSQ cell lines.

**Figure 12 cancers-11-00258-f012:**
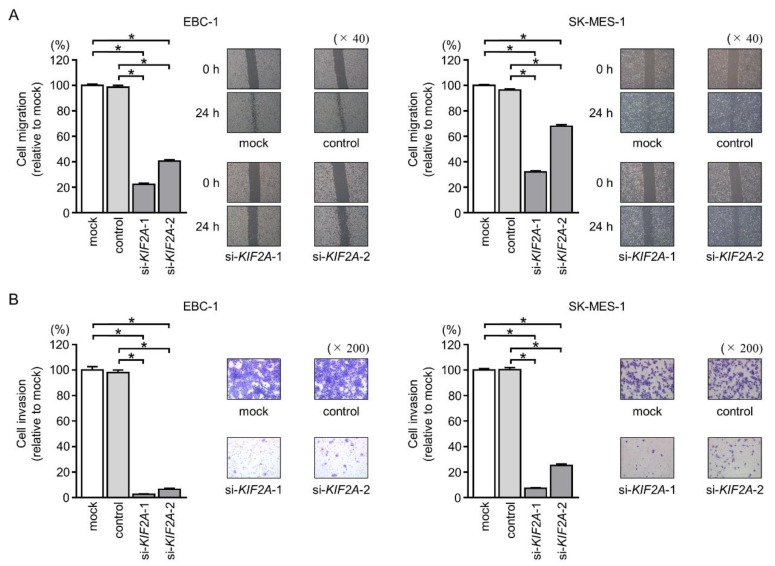
Effects of *KIF2A* silencing on cell migration and invasive abilities in LUSQ cells. (**A**) Cell migration was measured by wound healing assays (* *p* < 0.001). (**B**) Cell invasion was determined by Matrigel invasion assays (* *p* < 0.001).

**Table 1 cancers-11-00258-t001:** Characteristics of lung cancer and noncancerous cases.

**A. Characteristics of Lung Cancer Cases**		
**Total number**	**30**	
Median age (range)	71 (50–88)	
Sex	n	(%)
Male	29	(96.7)
Female	1	(3.3)
Pathological stage		
IA	5	(16.7)
IB	9	(30.0)
IIA	2	(6.7)
IIB	6	(20.0)
IIIA	7	(23.3)
IIIB	1	(3.3)
**B. Characteristics of noncancerous tissues**		
Total number	20	
Median age (range)	70.5 (50–88)	
Sex	n	
Male	20	
Female	0	

The pathological stage of lung cancer was classified according to Lung Cancer TNM classification, 7th Edition.

**Table 2 cancers-11-00258-t002:** Putative target genes regulated by *miR**-451a* in LUSQ cells.

Gene ID	Gene Symbol	Description	*miR-451a* Transfectant Log_2_ Ratio	*miR-451a* Target Site		GSE19188 Log FC	TCGA Database5-y OS *p*-Value
				Conserved Site	Poorly Conserved Site		
3796	*KIF2A*	kinesin family member 2A	−0.717	0	1	1.01	0.043
23200	*ATP11B*	ATPase phospholipid transporting 11B	−0.737	0	1	1.38	0.081
83990	*BRIP1*	BRCA1 interacting protein C-terminal helicase 1	−0.722	0	1	2.05	0.123
25769	*SLC24A2*	solute carrier family 24 member 2	−0.734	0	1	1.01	0.147
57405	*SPC25*	SPC25, NDC80 kinetochore complex component	−1.500	0	1	2.42	0.263
4282	*MIF*	macrophage migration inhibitory factor	−1.144	1	0	1.58	0.357
84951	*TNS4*	tensin 4	−0.592	0	1	2.56	0.494
1362	*CPD*	carboxypeptidase D	−1.233	0	1	1.07	0.598
23516	*SLC39A14*	solute carrier family 39 member 14	−1.190	0	1	1.01	0.607
5933	*RBL1*	RB transcriptional corepressor like 1	−0.922	0	1	1.08	0.720
81796	*SLCO5A1*	solute carrier organic anion transporter family member 5A1	−0.795	0	1	1.23	0.732
9699	*RIMS2*	regulating synaptic membrane exocytosis 2	−0.515	0	1	1.98	0.815
64067	*NPAS3*	neuronal PAS domain protein 3	−1.005	0	1	1.23	0.833
2668	*GDNF*	glial cell derived neurotrophic factor	−1.289	0	1	1.01	0.866
23657	*SLC7A11*	solute carrier family 7 member 11	−0.594	0	1	2.01	0.878

Lower and upper percentiles of The Cancer Genome Atlas (TCGA) database were both 33. GSE: Gene Expression Omnibus dataset results; FC: fold change; OS: overall survival.

**Table 3 cancers-11-00258-t003:** Characteristics and immunohistochemical status of patients in tissue microarray analysis.

A. Immunohistochemical status and characteristics of LUSQ cases							
**Patient No.**	**Grade**	**T**	**N**	**M**	**Pathological** **Stage**	**Immunohistochemical** **Staining Intensity**	**Immunohistochemical** **Staining Extensity**
23	2	2	1	0	IIB	(+)	(+++)
24	2	2	0	0	IB	(+++)	(+++)
25	2	1	0	0	IA	(+++)	(+++)
26	1	2	1	0	IIB	(++)	(+++)
27	2	1	0	0	IA	(++)	(+++)
28	1	3	0	0	IIB	(++)	(+++)
29	1	2	0	0	IB	(+++)	(+++)
30	2	2	0	0	IB	(+++)	(+++)
31	3	2	0	0	IB	(++)	(+++)
32	3	2	1	0	IIB	(+)	(+++)
33	3	2	0	0	IB	(++)	(+++)
34	3	2	1	0	IIB	(++)	(+++)
35	2	3	1	0	IIIA	(++)	(+++)
36	3	2	1	0	IIA	(++)	(+++)
37	3	3	0	0	IIB	(++)	(+++)
38	3	2	0	0	IB	(+++)	(+++)
39	3	2	1	0	IIB	(+++)	(+++)
40	3	2	0	0	IB	(+++)	(+++)
41	2-3	3	0	0	IIB	(++)	(+++)
42	3	1	2	0	IIIA	(+)	(+++)
43	3	2	0	0	IB	(++)	(+++)
44	3	2	0	0	IB	(++)	(+++)
B. Immunohistochemical status of noncancerous cases							
**Patient No.**						**Immunohistochemical** **Staining Intensity**	**Immunohistochemical** **Staining Extensity**
69						(++)	(++)
70						(++)	(+++)
71						(+)	(++)
72						(+)	(++)
73						(+)	(++)
74						(++)	(+++)
75						(+)	(++)
76						(++)	(++)
77						(+)	(++)
78						(+)	(+)
79						(++)	(++)
80						(++)	(+++)

The pathological stages of lung cancer were classified according to Lung Cancer TNM classification, 7th Edition.

**Table 4 cancers-11-00258-t004:** Significantly enriched annotations regulated by *KIF2A* in LUSQ cells.

No. of Genes	*p*-Value	Annotations
10	5.59 × 10^−12^	(KEGG) 04110: Cell cycle
3	5.71 × 10^−5^	(KEGG) 04115: p53 signaling pathway
-	-	(KEGG) 04110: Cell cycle
4	7.56 × 10^−5^	(KEGG) 04115: p53 signaling pathway
3	1.36 × 10^−4^	(KEGG) 03030: DNA replication
5	1.68 × 10^−3^	(KEGG) 05200: Pathways in cancer

**Table 5 cancers-11-00258-t005:** Downstream genes regulated by *KIF2A* among significantly enriched pathways in LUSQ cells.

Gene Symbol	Description	si-*KIF2A* Transfectant Log_2_ Ratio	GSE19188 Log FC
Cell cycle			
*CCNE2*	cyclin E2	−0.730	2.04
*MCM4*	minichromosome maintenance complex component 4	−0.925	3.13
*CCNA2*	cyclin A2	−0.563	3.25
*CCNE1*	cyclin E1	−1.099	2.23
*CHEK1*	checkpoint kinase 1	−0.586	2.76
*ORC6*	origin recognition complex subunit 6	−0.565	2.26
*CDC6*	cell division cycle 6	−0.529	3.31
*ORC1*	origin recognition complex subunit 1	−0.600	2.72
*CDC45*	cell division cycle 45	−0.524	3.83
*MCM2*	minichromosome maintenance complex component 2	−0.703	2.41
p53 signaling pathway, cell cycle			
*CCNE2*	cyclin E2	−0.730	2.04
*CCNE1*	cyclin E1	−1.099	2.23
*CHECK1*	checkpoint kinase 1	−0.586	2.76
p53 signaling pathway			
*CCNE2*	cyclin E2	−0.730	2.04
*CCNE1*	cyclin E1	−1.099	2.23
*CHEK1*	checkpoint kinase 1	−0.586	2.76
*RRM2*	ribonucleotide reductase regulatory subunit M2	−0.578	3.00
DNA replication			
*MCM4*	minichromosome maintenance complex component 4	−0.925	3.13
*MCM2*	minichromosome maintenance complex component 2	−0.703	2.41
*RFC4*	replication factor C subunit 4	−0.688	2.00
Pathway in cancer			
*MMP9*	matrix metallopeptidase 9	−0.531	2.04
*CCNE2*	cyclin E2	−0.730	2.04
*CBLC*	Cbl proto-oncogene C	−0.628	3.04
*CCNE1*	cyclin E1	−1.099	2.23
*RAD51*	RAD51 recombinase	−0.594	2.09

GSE: Gene Expression Omnibus dataset results; FC: fold change.

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
