# Peer review of "Regulation of KIF2A by Antitumor miR-451a Inhibits Cancer Cell Aggressiveness Features in Lung Squamous Cell Carcinoma"

_cancers, 2019, doi:10.3390/cancers11020258_

Reviewer 1 Report

This is a very nice study to explore the functional roles of miR-451a and to identify its targeting of oncogenic genes in LUSQ cells. The results clearly showed that low expression of miR-451a was significantly associated with poor prognosis of LUSQ patients, and by targeting kinesin family member 2A (KIF2A). I have a few comments below.

Is it possible to compare the expression level of miR-451a as well as KIF2A from normal tissue and LUSQ tissues in patients with lung cancer, instead of comparing them between patients with LUSQ and noncancerous cases?

It might be great if the results from this study can be confirmed from other clinical cases.

I am wondering why only one female patient diagnosed with LUSQ, how about the possibility that the results from this study can be applicable in female patients.

Author Response

Cancers Manuscript ID: cancers-444582

On behalf of my co-authors, I would like to revise an original research article entitled, “Regulation of KIF2A by antitumor miR-451a inhibits cancer cell aggressiveness features in lung squamous cell carcinoma” for consideration for publication in Cancers, Special Issue: MicroRNA-Associated Cancer Metastasis.

Reviewer #1:

Comment-1: Is it possible to compare the expression level of miR-451a as well as KIF2A from normal tissue and LUSQ tissues in patients with lung cancer, instead of comparing them between patients with LUSQ and noncancerous cases?
Response: I am correcting that there was ambiguity in the description about clinical samples. I will change the description of the Table 1 as follows; “noncancerous case” to “noncancerous tissues”.

I added the following sentence in Material and Method; “Lung cancer samples and noncancerous tissues were obtained from the lung specimens resected by thoracic surgery for LUSQ.”.

Comment-2: It might be great if the results from this study can be confirmed from other clinical cases.

Response: As suggested by the reviewer’s comment, I investigated the clinical significance of miR-451a and KIF2A expression in other types of cancers by using TCGA database. Based on the analysis results, I added Figure S3 and following sentences in results section; “In addition, TCGA database analyses showed that low expression of miR-451a was associated with poor prognosis in patients with renal papillary cell carcinoma and renal clear cell carcinoma.” and “In addition, TCGA database analyses showed that high expression of KIF2A was associated with poor prognosis in patients with renal papillary cell carcinoma and hepatocellular carcinoma.”.

Comment-3: I am wondering why only one female patient diagnosed with LUSQ, how about the possibility that the results from this study can be applicable in female patients.

Response: It is a fact that male and female are biased about the analyzed specimen. This is one of the clinical features of LUSQ. In TCGA database analyses of both miR-451a and KIF2A expression, one quarter of the total is female patients. Female patients with LUSQ was selected from the TCGA database and analyzed clinical significance of expression of miR-451a and KIF2A. However, we couldn’t obtain additional positive results. The personal figure for these analyses is shown only to reviewers. We will not mention that results in the manuscript. I appreciate your understanding.

Figure P1 is in the attached file.                                            

Figure P1. Kaplan-Meier analyses of miR-451a and KIF2A expression among female patients with LUSQ. Kaplan-Meier curve of 5-year overall survival according to miR-451a and KIF2A expression among female patients with LUSQ in TCGA database (p = 0.104 and p = 0.551, respectively). Patients were divided into high (red) and low (blue) expression groups.

Our data are the first to report these findings in LUSQ. We believe that this article contributes significantly to our understanding of the molecular mechanisms of LUSQ pathogenesis. We appreciate your consideration of this manuscript for publication in Cancers, Special Issue: MicroRNA-Associated Cancer Metastasis.

Sincerely yours,

Naohiko Seki, Ph.D.

Department of Functional Genomics

Chiba  University Graduate  School of Medicine

1-8-1 Inohana, Chuo-ku,

Chiba 260-8670,Japan

Phone: +81-43-226-2971

Fax: +81-43-227-3442

Reviewer 2 Report

Submitted manuscript titled “Regulation of KIF2A by antitumor miR-451a inhibits cancer cell aggressiveness features in lung squamous cell carcinoma” by Uchida A, et al. performed clinicopathological and cell biological studies to clarify the role of miR-451a in the development of LUSQ (Lung Squamous Cell Carcinoma). They observed that the miR-451a levels in LUSQ tissues and cells were significantly decreased. The analyses of TCGA database showed that the low expression of mir-451a correlated with poor prognosis of LUSQ cases. Ectopic expression of miR-451a induced apoptosis of LUSQ cells and restricted the migration and invasion of LUSQ cells. KIF2A was validated as a target gene of miR-451a, by observing the down-regulation of KIF2A expression in LUSQ cells with miR-451a transfection and by the confirmation of direct binding of mir-451a to 3’UTR of KIF2A mRNA with luciferase assays. Moreover, KIF2A was overexpressed in LUSQ tissue and its knockdown in LUSQ cells resulted in apoptosis of LUSQ cells and restricted the migration and invasion of LUSQ cells.

Overall, the research is well-organized and provides some useful information for the potential role of miR-451a in LUSQ development. However, the manuscript requires some revision as listed below to attain sufficient quality for the publication in “Cancers”.

Major points

1.  Since platinum-based chemotherapy is performed for inoperable cases of LUSQ, as the authors point out in the introduction, it is of interest to observe combinatorial effect of cisplatin and mir-451a induction or KIF2A silencing in LUSQ cells.

2.  What was the correlation between miR-451a expression and KIF2A expression in the clinical samples analyzed in the study?

3. Tobacco smoking is a major etiology of LUSQ. How about the correlation between tobacco smoking (smoking index) and miR-451a and KIF2A status in LUSQ cases?

4. The authors are advised to discuss the molecular mechanism for the down-regulation of miR-451a in LUSQ.

5. The number of the LUSQ tissues was limited (n=30) for the mir-451a expression analysis. The authors are advised to strengthen the discussion by referring the publication below which analyzed 73 LUSQ clinical samples. . Goto A, et al. The low expression of miR-451 predicts a worse prognosis in non-small cell lung cancer cases. PLoS One. 2017 Jul 12;12(7):e0181270. doi: 10.1371/journal.pone.0181270. eCollection 2017.

Author Response

Cancers Manuscript ID: cancers-444582

On behalf of my co-authors, I would like to revise an original research article entitled, “Regulation of KIF2A by antitumor miR-451a inhibits cancer cell aggressiveness features in lung squamous cell carcinoma” for consideration for publication in Cancers, Special Issue: MicroRNA-Associated Cancer Metastasis.

Reviewer #2:

Comment-1: Since platinum-based chemotherapy is performed for inoperable cases of LUSQ, as the authors point out in the introduction, it is of interest to observe combinatorial effect of cisplatin and mir-451a induction or KIF2A silencing in LUSQ cells.

Response: I would like to express my deep gratitude to reviewer’s comment. Currently, we are searching for microRNAs that enhance the anticancer effect of cisplatin in LUSQ cells. For this analysis, it is also necessary to establish a cisplatin-resistant LUSQ cell lines. In this thesis, we do not indicate data on the involvement of miR-451a/KIF2A with anticancer drugs. I appreciate your understanding.

Comment-2: What was the correlation between miR-451a expression and KIF2A expression in the clinical samples analyzed in the study?

Response: As suggested by the reviewer’s comment, I investigated the correlation between miR-451a/KIF2A expressions in LUSQ patients by using TCGA database. Based on the analysis results, I added Figure S6 and following sentences in Results; “We also investigated the correlation between miR-451a and KIF2A expression in LUSQ patients. TCGA database analyses showed that a negative correlation was detected between miR-451a and KIF2A expression in LUSQ patients (r = -0.180 and p = 0.010; Figure S6).”.

Comment-3: Tobacco smoking is a major etiology of LUSQ. How about the correlation between tobacco smoking (smoking index) and miR-451a and KIF2A status in LUSQ cases?

Response: I am grateful for your kind advice. According to the reviewer’s comment, I examined the description of the tobacco smoking in LUSQ patients in TCGA database. However, it was not possible to obtain accurate and sufficient data to analyze smoking index in LUSQ patients. Unfortunately, this analysis was impossible this time. I appreciate your understanding.

Comment-4: The authors are advised to discuss the molecular mechanism for the down-regulation of miR-451a in LUSQ.

Response: The reviewer's point it out is important to this paper. We cited the previous studies and added the following description about molecular mechanism of downregulation of miR-451a in cancer cells.

“Elucidation of molecular mechanisms of aberrantly expressed miRNAs in cancer cells is an important issue for cancer research. Previous study showed that expression of miR-451a was significantly recovered by treatment with 5-aza-2’-deoxycitidine or sodium plenylbutyrate in NSCLC cells. (Wang et al, Oncogene, 2011). These results indicated that DNA hypermethylation was caused to downregulation of miR-451a in NSCLC. Recent study of prostate cancer showed that HP1γ was upregulated by oncogenic c-MYC and HP1γ suppressed to expression of miR-451a in prostate cancer cells (Chang et al, Oncogene, 2018). Further investigation of the molecular mechanism of downregulation of miR-451a in LUSQ cells is indispensable.”

Comment-5: The number of the LUSQ tissues was limited (n=30) for the mir-451a expression analysis. The authors are advised to strengthen the discussion by referring the publication below which analyzed 73 LUSQ clinical samples. Goto A, et al. The low expression of miR-451 predicts a worse prognosis in non-small cell lung cancer cases. PLoS One. 2017 Jul 12;12(7):e0181270. doi: 10.1371/journal.pone.0181270. eCollection 2017.

Response: I am grateful for your kind comment. We cited the previous studies and added the following description about clinical significance of miR-451a expression in non-small cell lung cancer.

“Furthermore, downregulation of miR-451 was detected in NSCLC tissues and its expression was an independent predictor of prognosis of NSCLC, such as advanced disease stage and metastasis (Goto et al, PLos One, 2017). Interestingly, ectopic expression of miR-451 suppressed cell proliferation, migration and activation of AKT through targeting MIF in NSCLC cells (Goto et al, PLos One, 2017).”

 Our data are the first to report these findings in LUSQ. We believe that this article contributes significantly to our understanding of the molecular mechanisms of LUSQ pathogenesis. We appreciate your consideration of this manuscript for publication in Cancers, Special Issue: MicroRNA-Associated Cancer Metastasis.

Sincerely yours,

Naohiko Seki, Ph.D.

Department of Functional Genomics

Chiba  University Graduate  School of Medicine

1-8-1 Inohana, Chuo-ku,

Chiba 260-8670,Japan

Phone: +81-43-226-2971

Fax: +81-43-227-3442

This manuscript is a resubmission of an earlier submission. The following is a list of the peer review reports and author responses from that submission.